# Analysis and Design of Operating Parameters of Floor-Standing Jujube Pickup Device Based on Discrete Element Method

**Lun Zhou** [1,2], **Jingbin Li** [1,2,*], **Longpeng Ding** [1,2], **Huizhe Ding** [1,2] **and Junpeng Liang** [1,2]

1 College of Mechanical and Electrical Engineering, Shihezi University, Shihezi 832000, China
2 Xinjiang Production and Construction Corps, Key Laboratory of Modern Agricultural Machinery, Shihezi 832000, China
* Correspondence: lijingbin@shzu.edu.cn

**Abstract:** In view of the problem whereby the floor-standing jujube pickup device designed by the research group has a large resistance when the comb teeth (CT) enter the soil, the strip brush can easily bend, and the operation effect is poor. In this paper, the structural parameters of the CT and bar brush in the mechanical floor date-picking device are determined by theoretical analysis. EDEM software was used to establish the discrete element simulation model of the floor-standing jujube soil–jujube pickup device. According to the simulation results, it is determined that the angle of the CT entering the soil (ACT) can change freely within 30–33° and meet the operation requirements. Through the single-factor test on the pickup rate of jujube and the soil hilling quantity, it is determined that the value range of the CT working speed (CWS) is 0.2–0.4 m/s, and the value range of the CT entering the soil (DCS) is 5–45 mm. Simulation and field verification tests were carried out on the determined operation parameter range. It was found that when the CWS was 0.2 m/s and the DCS was 5 mm, the pickup rates in the simulation test and the field verification test were 84.17% and 91.23%, respectively, and the relative error was the largest but not more than 8%. The result shows that the operation parameters and range determined by the discrete element method were reliable. This study is expected to provide the working parameter basis for the subsequent design of a floor-standing jujube pickup device.

**Keywords:** jujube; machine design; pickup device; EDEM; simulation



## 1. Introduction

Jujubes are warm and sweet, with high nutritional value, and are harvested at the beginning of November each year. Xinjiang is the main jujube-producing area in China [1–3]. The Xinjiang jujube industry is essential for promoting farmers' income and adjusting the economic structure of the market [4,5]. While jujube is developing towards industrialization and large-scale production, the harvesting problem of jujube has become a key factor that restricts the development of the jujube industry [6]. In recent years, the mainstream jujube-harvesting method in Xinjiang has been the manual picking of fallen jujubes, so mechanized picking rather than human labor is the main research direction at present. Therefore, based on the working principle of the comb teeth (CT) entering the soil and rotating strip brush picking, the group designed a floor-standing jujube pickup device (hereinafter referred to as the pickup device). Field tests have shown that the pickup device works efficiently, with a pickup rate of more than 90%. However, the operational parameters that appear when working are not matched, and it is unreasonable and complicated to adjust the parameters, which makes the structure of the device complicated and the performance not stable enough, resulting in the jujubes being easily buried by the soil, and the picking operation being seriously constrained by the soil. Therefore, it is of great significance to explore the most appropriate operation parameters to improve the pickup performance of the lifting device and the disturbance law of the device on the soil.

The discrete element method (DEM) is an excellent numerical simulation and analysis method in the agricultural field [7–9], with important guiding significance for the interaction between soil and tools. Wang Xuezhen and Hang Chengguang et al. [10,11] used the discrete element method to simulate the operation of a subsoiling shovel in soil and explore the disturbance characteristics of the soil. Zhang Zhiyuan et al. [12] used the DEM method to explore the soil hilling quantity condition during the operation of an arc-shaped nail-tooth roller-type recovery machine for sowing layer residual film and determined the main working parameters affecting the operation of the device and the value range of each parameter. Zhao Shuhong et al. [13] established a discrete element model of subsoiling shovel soil straw stubble by using EDEM to explore the movement law of straw on the ridge under the action of the subsoiling shovel. Comparing the simulation test with the field test, the maximum error was 11.06% and the minimum error was 0.16%. Hou Jialin et al. [14] compared the screening combined scallion thinning device with the original scallion digging and shaking soil thinning device via the discrete element method and a field test and explored the disturbance characteristics of the two devices on the soil under the same operating conditions. Zhao Shuhong [15] et al. used the EDEM software to simulate and analyze the action mechanisms of different types of stubble-breaker fertilizer shovel devices on the soil and designed a segmented corn seeder. In summary, the discrete element method can successfully simulate the interaction characteristics of soil contacting parts and soil and has a certain reference value for exploring the operating speed, operating depth, soil resistance, and other parameters of machines.

Based on the previous research work, this paper determines the structural parameters and operating parameters of each working part of the floor-standing jujube pickup device by theoretical analysis and pretests. The EDEM software was used to build the floor-standing jujube pickup device, soil and jujube simulation model, and the influence of the angle of the CT entering the soil (ACT) on the jujube pickup rate and the soil hilling quantity was analyzed through the simulation test, while the free variation range of the ACT was determined. Single-factor tests were carried out on the CT working speed (CWS) and the depth of the CT entering the soil (DCS) to determine the operating parameter range of each factor. The accuracy of the simulation test was verified by comparing the field test with the simulation test. This study can provide the operation parameter basis and technical support for the development of a floor-standing jujube pickup device to a certain extent.

## 2. Materials and Methods

In order to determine the structural parameters of the strip brush and comb teeth (CT), which are the core components of the pickup device, and determine the influencing factors that interfere with the CT operation. This paper theoretically analyzes the interaction between brush and jujube and the interaction between sandy soil and CT.

### 2.1. Pickup Device Operation Object

During the harvesting period, most jujubes will fall under the influence of wind, rain, and other external factors due to the low connection force between jujubes and jujube branches. Therefore, the current harvesting method of Xinjiang jujube mainly consists of knocking down the remaining jujubes on the tree via human labor, and then collecting the fallen jujubes into strips (manual strip collection—MCS) and finally picking up the jujubes by hand to complete the harvesting operation [5] (Figure 1). The floor-standing jujube pickup device studied in this paper is applicable to fallen jujubes after MCS in dwarf dense jujube orchards, where the width of the jujube after MCS is less than 1 m; the row spacing of the jujube orchard is 3–4 m, and the soil type is sandy soil.

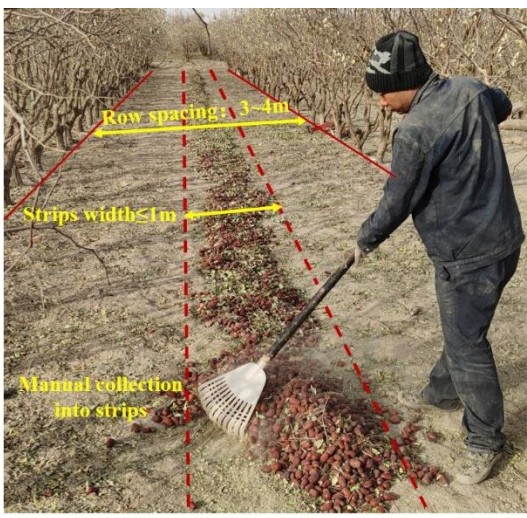

**Figure 1.** Manual strip collection—MCS.

*2.2. Structure and Working Principle*

To achieve the intended design functions, the floor-standing jujube pickup device is based on the mechanical floor-standing jujube pickup machine developed by the research group. The structure of the whole machine is shown in Figure 2a. The machine is mainly composed of three parts: a working part, a transmission part, and a jujube storage part. The machine can complete jujube picking, transportation, impurity removal, and fruit collection at one time. As the core structure of the machine, the pickup device consists of a CT, picking side plate, strip brush, strip brush roller shafts, hydraulic cylinders, profiling wheels, profiling adjusting rods, rolling pulleys, lifting rods, and other components. The schematic diagram of the jujube mechanism is shown below (Figure 2b).

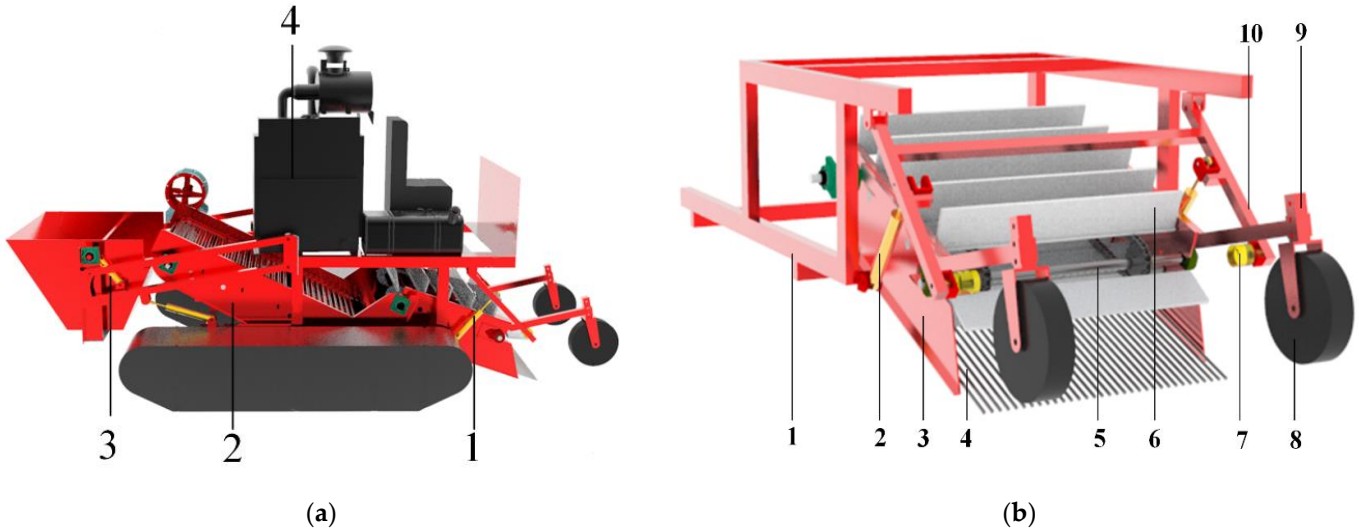

(**a**)  (**b**)

**Figure 2.** (**a**) The whole machine structure. (1) Pickup device. (2) Conveying device. (3) Jujube storage device. (4) Drive system. (**b**) Pickup device three-dimensional schematic. (1) Rack. (2) Device lifting cylinder. (3) Pickup side plate. (4) CT. (5) Brush roller. (6) Strip brush. (7) Rolling pulley. (8) Profile wheel. (9) Profiling adjusting rod. (10) Lifting rod.

Working Principles of the Pickup Device

Before the operation of the device, the pickup device is dropped by the lifting rod, and the operating parameters of the device are adjusted by adjusting the position of the profiling rod bolt. During the working process, the machine moves towards the working area, and the dates, sand, date branches, and date leaves are left behind the MCS piled up

by the CT. The rotation of the strip brush causes the materials to accumulate and climb along the CT, where they then move to the conveying device under the force. At the same time, impurities, such as sand, jujube branches, and jujube leaves can be filtered through the gap between the CT during movement. Thus far, the pickup device has completed the pickup operation of the fallen jujubes.

### 2.3. Design of Key Components

2.3.1. Working Principles of the Pickup Device

In order to determine the position parameters of each component of the lifting mechanism, the lifting principle diagram shown in Figure 3 is analyzed. Among them, *AE* and $AE_1$ are the limit positions of the lifting rod. The lifting rod controls the rolling pulley to act on the extension end of the pickup side plate under the action of the hydraulic cylinder, so as to control the lifting and lowering of the device. In order to determine the basic structural parameters of the lifting mechanism, we can utilize the geometric relationship shown in Figure 3.

$$\frac{L_{EE_1}}{L_{CD}} = \frac{L_{AC}}{L_{AE}} = a \tag{1}$$

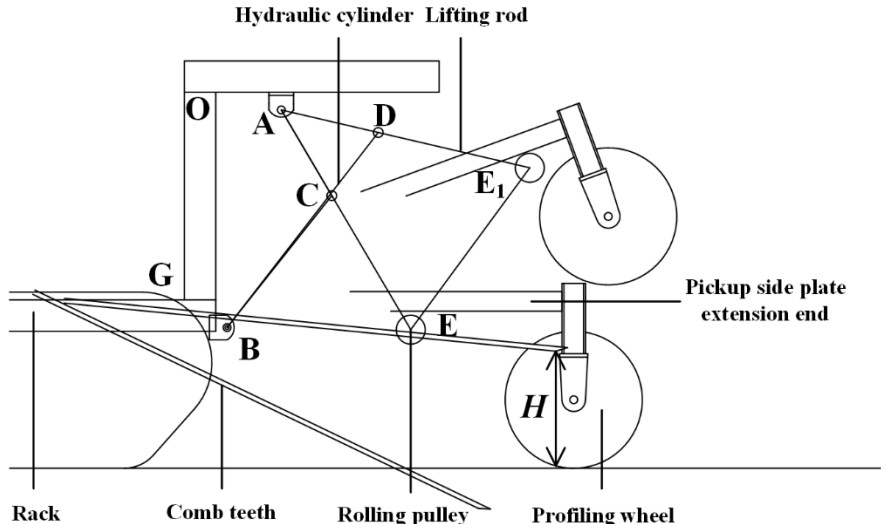

**Figure 3.** Schematic diagram of lifting principle.

Since the value of $\angle DBC$ is very small, the straight-line *BC* and *BD* can be regarded as approximately collinear. The $L_{CD}$ should be the effective working length of the selected hydraulic cylinder, and *H* is the maximum distance from the CT to the ground after lifting. According to the actual operation requirements, we set the maximum distance between the CT and the ground as *H* = 200 mm, the effective working length of the selected hydraulic cylinder is 200 mm, and the lifting rod length is 650 mm. Thus, it is determined that the distance between the hinge point A and the frame *OG* is 167.17 mm. According to the parameters of the pickup device and the geometric relationship shown in the figure, $L_{EE1}$ is 521.85 mm, and the ratio a = 2.61 and $L_{AC}$ = 250 mm can be obtained from Equation (1). It can be seen that the distance between the rod end C of the hydraulic cylinder and the hinge A is 250 mm.

2.3.2. Analysis of Factors Affecting Strip Brush Picking of Jujubes and Design of Related Structural Parameters

After the jujube is picked up by the CT, the rotary bar brush sweeps up the jujube, and the picked jujubes move backward with the brush. In order to explore the key factors affecting the picking operation of the strip brush, we reduce the bending phenomenon of the strip brush in the picking process and improve the harvesting effect. The interaction between the strip brush and the jujubes was simplified by taking a single strip brush as the

unit and treating the jujube as an ellipse. Since one end of the strip brush is fixed to the chain side, point A, where the strip brush is fixed to the chain side, can be simplified as a fixed-end constraint. From the force analysis of the strip brush in Figure 4, we can obtain the following:

$$
\begin{cases}
F_f = \mu_1 F_H \\
F_{A1} = F_H \cos\rho_1 + \mu_1 F_H \sin\rho_1 \\
F_{A2} = \mu_1 F_H \cos\rho_1 - F_H \sin\rho_1
\end{cases}
\tag{2}
$$

where $Ff$ is the friction force of jujube on the strip brush, N; $F_{A1}$ is the vertical component force of jujube on top A of the strip brush, N; $F_{A2}$ is the horizontal component force of jujube on top A of the strip brush, N; $F_H$ is the supporting force of the jujube on the strip brush at point H, N; $\mu_1$ is the friction coefficient between the jujube and strip brush; $\rho_1$ is the included angle.

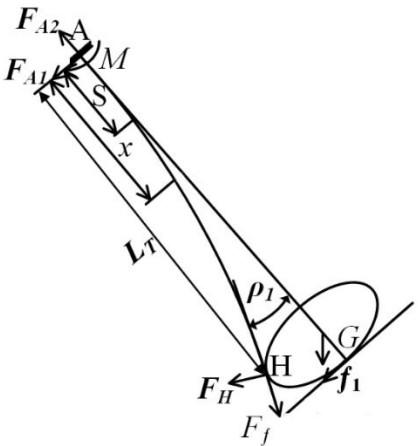

**Figure 4.** Sketch of the interaction between strip brush and jujube.

Then, under the interaction of force, the bending moment generated by the strip brush at any $x$ from the top A is $M_0$. Assuming that bending occurs at $S$, the bending moment $M$ on the section at $S$ is:

$$
\begin{cases}
M_0 = (F_H \cos\rho_1 + \mu_1 F_H \sin\rho_1)(L_T - x) \\
M = (F_H \cos\rho_1 + \mu_1 F_H \sin\rho_1)(L_T - S)
\end{cases}
\tag{3}
$$

where $L_T$ is approximately the strip brush length, mm.

Among the parameters, the stresses caused by axial force $F_{A2}$ and bending moment $M$ are, respectively,

$$
\begin{cases}
\sigma_{A2} = \dfrac{4F_{A2}}{\pi^2 D^2} \\[2mm]
\sigma_M = \dfrac{M}{W}
\end{cases}
\tag{4}
$$

where $D$ is the cross-sectional diameter of the strip brush, mm; $W$ is the bending section coefficient of the strip brush, mm$^3$.

According to the stress superposition, the comprehensive normal stress $\sigma_Z$ of the section at S is composed of the axial force stress $\sigma_{A2}$ and the bending normal stress $\sigma_M$ [16]; then, the comprehensive normal stress $\sigma_Z$ of the section $S$ is:

$$
\sigma_Z = \frac{4(\mu F_H \cos\rho_1 - F_H \sin\rho_1)}{\pi D^2} + \frac{(F_H \cos\rho_1 + \mu_1 F_H \sin\rho_1)(L_T - S)}{W}
\tag{5}
$$

From Equations (2) and (5), it is clear that $\sigma_Z$ will increase with the increase of the support force. The greater the supporting force, the easier it is to cause the bending of the strip brush. At the same time, it can be seen from the interaction of forces in Figure 4 that if the jujubes become excessively piled up, the gravitational component of the jujube and the friction force between the jujube and the CT will become larger. Then, the support

force and friction of the jujube on the strip brush will increase, so that the normal stress at the S section of the strip brush will increase, and the strip brush will be more likely to bend at the S section, which will affect the strip brush operation. It can also be seen from Equation (5) that, in practice, if a single strip brush is replaced with a strip brush plate of a certain thickness, the actual cross-sectional diameter of the strip brush increases. Then, the comprehensive normal stress will become small, and the strip brush will not easily bend, but it can easily damage the jujube. Therefore, in order to determine the material and operating parameters of the strip brush [17], the approximate differential equation of the deflection curve is obtained.

$$
\begin{cases}
\dfrac{d^2w}{dx^2} = \dfrac{M}{EI} \\
EIw'' = M = (F_H\cos\rho_1 + \mu_1 F_H\sin\rho_1)(L_T - x)
\end{cases}
\tag{6}
$$

If $(F_H\cos\rho_1 + \mu_1 F_H\sin\rho_1)$ is $F$, then

$$
EIw = \frac{F}{6}x^3 - \frac{FS}{2}x^2 + C
\tag{7}
$$

where w is the deflection value; $I$ is the moment of inertia of the cross-section of the strip brush to its center axis, $mm^4$.

At the fixed end A of the bristles, the deflection value is 0, and when $x = 0$, $w_A = 0$ is represented by Equations (6) and (7).

$$
C = EIw_A = 0
\tag{8}
$$

When $x = L_T$, the deflection of section $x$ is:

$$
W_x = -\frac{FL_T{}^3}{3EI}
\tag{9}
$$

From Equation (9), it can be seen that the larger the elastic modulus of bristles, the smaller the deflection value and the smaller the elastic deformation, and the greater the stiffness of the strip brush. Although it is not easy to fold the brush when the jujubes accumulate too much, it will increase the damage of the jujubes. The longer the bristles, the greater the deflection. Although the damage to the jujube will be reduced, the contact position between the strip brush and the jujube will be advanced, resulting in the included angle $\rho_1$ becoming large, causing the strip brush to bend easily. According to the pretest conducted by the research group, the material of the strip brush is determined to be PVC with an elastic modulus of 3.14 GPa. Formed into a length of 150 mm, a 5 mm thickness of the strip brush plate can meet the requirements for picking up jujubes.

### 2.3.3. CT Structure Design and Force Analysis of the CT in the Soil

The installation position of the bar brush rollers relative to the CT and the length of the CT are the key factors affecting the picking of jujubes. Among them, the strip brush roller shaft can be finely adjusted through the bearing hole on the pickup side plate, but the CT length cannot be easily changed during operation. Therefore, the appropriate CT length is an important structural parameter of the pickup device. In addition, the force state of the soil-touching component CT of the picking device in the jujube orchard soil can directly affect the picking effect. Therefore, a theoretical analysis is needed to clarify the key factors affecting the working state of the CT in the jujube orchard.

### 2.3.4. CT Structure Parameter Design

In order to obtain a suitable comb length, it was assumed that jujubes, soil, residual branches, and fallen leaves (jujube-branch-soil mixture) accumulated at point A while the device advances under the drive of the power system (Figure 5). The jujube–branch-soil mixture then climbs from point A to end point B [18] at a speed $v_A$ and is in a stationary

state. At this moment, the speed along the CT surface is 0, and the jujube sand mixture begins to fall loose on both sides.

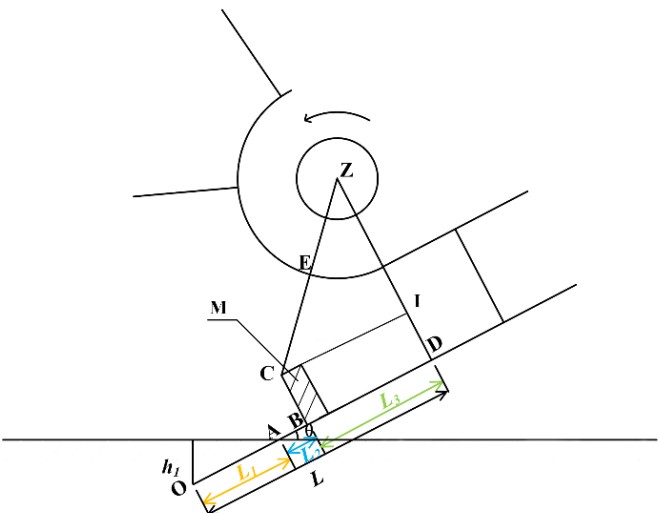

**Figure 5.** Schematic diagram of the mixture of dates and soil being shoveled up and climbed by the CT of the soil touching part of the picking device.

The distance $L_1$ between the CT end $O$ and the contact point A between the CT and the ground is:

$$L_1 = \frac{h_1}{\sin\theta} \tag{10}$$

where $\theta$ is the angle of the CT when working in the soil; $h_1$ is the depth of the CT when working in the soil, mm. According to the law of energy conservation, the kinetic energy of the jujube sand mixture is all used to overcome the work done by the friction force $W_f$ of the $L_2$ segment rising along the CT, and the gravitational potential energy $W_G$ of the jujube rising; there are:

$$\begin{cases} W_G = mgL_2\sin\theta \\ W_f = mgL_2\mu\cos\theta \end{cases} \tag{11}$$

where $m$ is the mass of the jujube-branch-soil mixture, kg; $\mu$ is the friction coefficient for the action between the jujube-branch-soil mixture and the working parts [19].

Therefore, the energy conservation equation of the $L_2$ segment is as follows:

$$\frac{mv_A{}^2}{2} = W_f + W_G \tag{12}$$

where $v_A$ is the velocity of the jujube-sand mixture at point A of the comb, m/s.

The end of the strip brush is in contact with the jujube-sand mixture at point C, and then, the end of the strip brush moves from point C to point D, where the height of the CB segment is the height of the jujube-sand mixture after the strip is gathered, and the distance of $L_3$ is as follows:

$$L_3{}^2 = L_{ZC}{}^2 - (L_{ZD} - L_{ID})^2 \tag{13}$$

Then, from Equations (10), (11), and (13), the distance L from the end O of the CT to the projection point D of the strip brush roller shaft axis Z on the comb tooth is known as:

$$L = \frac{h_1}{\sin\theta} + \frac{v_A{}^2}{2(\mu g\cos\theta + g\sin\theta)} + \sqrt{L_{ZC}{}^2 - (L_{ZC} - L_{BC})^2} \tag{14}$$

Among them, the average thickness $L_{BC}$ of the jujube sand mixture paved on the ground after MCS is 30 mm~150 mm [20]. According to the pretest, $v_A$ is generally slightly lower than the CWS. For the convenience of calculation, the CWS is used instead of $v_A$.

$\mu$ is taken as 0.035 [19], and the length $L_{EC}$ of the strip brush is 150 mm, as shown above. In actual operation, the jujube sand mixture scooped up by the back will give support to the jujube sand mixture that has already climbed, meaning that the jujube sand mixture will not fall back after climbing, so the jujube sand mixture is higher than the theoretical climbing. Therefore, the theoretical maximum value is taken for the DCS, which is set to 60 mm according to references [5,17]. The CWS is 0.6 m/s, the ACT is 30°, and the $L_{BC}$ is 90 mm. From Equation (14), $L$ is 261.5 mm, which is taken as 262 mm. The distance between the installation position of the brush roller axis Z and the end of the CT is 262 mm. Combined with the length of the pickup device, the total length of the CT is 1050 mm.

The CTs are selected from round steel with a diameter of 12 mm. The size of the spacing between the CT is controlled by the size dimensions of the jujube. The measured size dimensions of the jujubes were 19.23–41.53 mm, so the spacing dimension between the CT was determined to be 18 mm. The operating width of the pickup device is 1100 mm, so the total number of CT required for the pickup device is 37.

2.3.5. Force Analysis of CT in Working Conditions in Jujube Orchard Soil

According to the pretest, the pickup effect decreases significantly when the CTs are in poor working condition. Therefore, in order to determine the factors affecting the working state of the CT entering the soil and to improve the pickup effect, a force analysis of the operating process of the CT in the soil was performed (Figure 6a). Figure 6 shows the force state of the CT in the soil. By analyzing this force state, the balance equation of the CT in the forward direction can be obtained as [21].

$$\begin{cases} F_W = F_0\sin\theta + F_f\cos\delta + F_k\cos\theta \\ F_f = \mu_0 F_0 \end{cases} \tag{15}$$

where $F_W$ is the force required for the device to move forward, N; $F_0$ is the normal load on the CT, N; $\theta$ is the working angle of CT in soil CT, (°); $F_f$ is the friction force between sandy soil and CT, N; $\delta$ is the inclination angle of the failure surface before the jujube orchard soil block, (°); $\mu_0$ is the friction coefficient between sandy soil and CT; $F_K$ is the sandy soil cutting resistance, N.

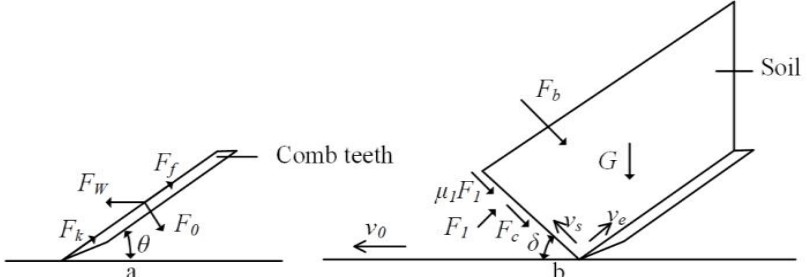

**Figure 6.** The force analysis diagram of CT in the working process of jujube orchard soil. (**a**) Force analysis of CT in soil working process, (**b**) The force of accumulated soil on CT.

Since there are no large stones or residual roots left in the sandy soil environment of the jujube orchard, the cutting resistance $F_K$ can be considered as 0. Therefore, during the advancement of the CT, CT will be subject to the stress of sand accumulated on the surface. (Figure 6b). The equilibrium state equation of the sandy soil in the horizontal and vertical directions is as follows:

$$\begin{cases} F_0(\sin\theta + \mu_0\cos\theta) - F_1(\sin\delta + \mu_1\cos\delta) - (F_c + F_b)\cos\theta = 0 \\ G - F_0(\cos\theta - \mu_0\cos\theta) - F_1(\cos\delta - \mu_1\cos\delta) + (F_c + F_b)\sin\delta = 0 \\ F_b = m\frac{dv}{dt} \end{cases} \tag{16}$$

where $\mu_1$ is the friction coefficient for sandy soil; $F_C$ is the cohesion of sandy soil, N; $F_b$ is the acceleration force of sandy soil, N; $G$ is the sandy soil gravity, N; $m$ is the mass of the accelerated sandy soil, kg; $v$ is the velocity of the sandy soil being accelerated, m/s.

When solving the acceleration force $F_b$ of sandy soil, according to Soehne's proof [21], if the periodic failure process is continuous, the total work required to generate the above acceleration will not change significantly, and an average constant force can be determined. Moreover, the sandy soil is stationary at $t = 0$.

In addition, the magnitude of each velocity vector is considered to form a closed triangle, so

$$\begin{cases} m = \frac{\rho}{g}bdtv_0 \\ \frac{dv}{dt} \gg \frac{Vv}{Vt} = \frac{v_s - 0}{t - 0} = \frac{v_s}{t} \\ v_s = v_0 \frac{\sin\theta}{\sin(\theta + \delta)} \end{cases} \tag{17}$$

where $\rho$ is the density of sandy soil, kg/m$^3$; $g$ is the acceleration of gravity, m/s$^2$; $b$ is the width of CT operation, mm; $d$ is the disturbance depth of sandy soil, m; $t$ is the time, s; $v_0$ is the working speed of the CT, m/s; $v_s$ is the velocity of sandy soil along the shear plane, m/s.

According to Equation (18), the solution of $F_b$ is as follows:

$$F_b = \frac{\rho}{g}bdv_0{}^2 \frac{\sin\theta}{\sin(\theta + \delta)} \tag{18}$$

During the operation of the machine, theoretically, the driving speed is kept constant, so the traction force $F_W$ and the sandy soil resistance $F_{W1}$ are a pair of equilibrium forces, which can be obtained from Equation (16)–(18):

$$F_{W1} = \frac{G + \frac{F_c + F_b}{\sin\delta + \mu_1\cos\delta}}{\frac{\cos\theta - \mu\sin\theta}{\sin\theta + \mu_0\cos\theta} + \frac{\cos\delta - \mu_1\sin\delta}{\sin\delta + \mu_1\cos\delta}} \tag{19}$$

From the above analysis, it can be seen that the working condition of the CT entering the soil is affected by the working width, the CWS, the DCS, the ACT, the inclination of the failure surface in front of the soil block, and the gravity of the soil. The front shear failure area of the soil block and the inclination angle of the front failure surface of the soil block $\delta$ depend on the angle of the comb teeth in the work of the jujube orchard soil $\theta$ [22,23]. Therefore, the factors obtained from the above analysis are the main reasons affecting the working status of CWS, DCS, and ACT.

### 2.4. Analysis of Pickup Device Operation Parameters Based on DEM

Simulation Model Establishment and Parameter Determination

In order to efficiently and quickly analyze the disturbance law of the main factors affecting the CT operation on the soil, determine the value range of the CT operation parameters. In the simulation test of the pickup device after the MCS, the simulation model consists mainly of sandy soil, jujube, and the pickup device. The modeling of the floor-standing jujube pickup device was performed in the 3D modeling software SolidWorks, as well as the simulation. Previous research has reported that the shape of the tested material has a greater impact on the simulation results, while the size of the tested material has a small impact on the simulation results [24–27]. After observing the appearance of jujube in its early stage, the representative appearance of jujube was selected, as shown in Figure 7 [6]. Using the Python programs, we can extract the boundary contour of the selected jujube, import the jujube contour into Adobe Photoshop, and use it to eliminate the ravines in the contour. After cutting along the edge of the image grid and importing it into CAD software [28], the outline of the jujube is described by the spline curve, and its outline is modeled using the SolidWorks software. The obtained jujube model is imported

into the EDEM software, and two jujube simulation models are established by using the particle filling function.

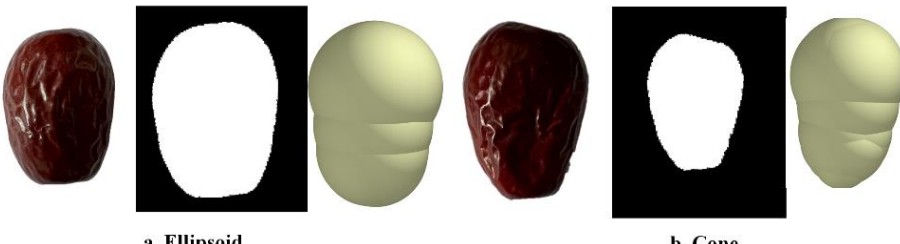

**a. Ellipsoid**                                        **b. Cone**

**Figure 7.** Schematic diagram of jujube simulation model.

The selection of a particle contact model is an important basis for ensuring the accuracy of the simulation results of the simulation analysis model. In order to simulate the interaction between soil and machines in the actual production process, this study set the soil particle radius to 3 mm, with reference to [17,29,30]. The soil in the working jujube orchard has low water content, good mobility [17,20], and no bonding effect with the working parts, so the Hertz Mindlin (No Slip) contact model in the EDEM software was used. In order to ensure the simulation effect and reduce the computation time, we set the sandy soil particle model as a circle. The particle factory generates sandy soil particles dynamically, and the particles settle and accumulate into sandy soil. To ensure the working range of the working parts, the simulation areas of the soil trough with a length of 2000 mm, a width of 600 mm, and a height of 150 mm were established in this study. At the same time, a jujube strip belt with a row spacing of 220 mm was established in the particle factory to replace the MCS. The model and simulated soil trough of the ground jujube pickup device are shown in Figure 8.

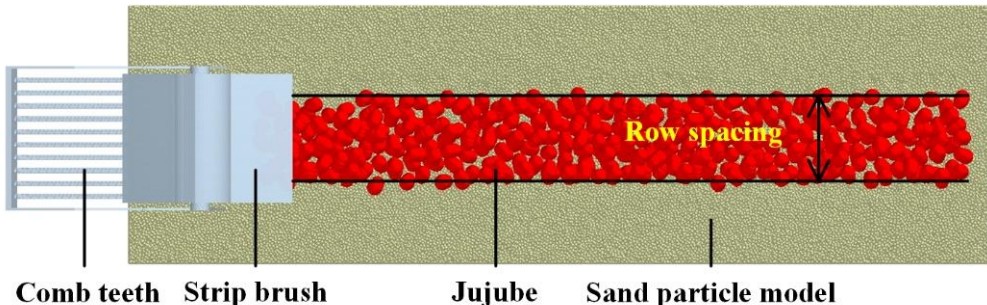

**Comb teeth  Strip brush        Jujube    Sand particle model**

**Figure 8.** Simulation test model diagram.

Since this paper mainly analyzes the pickup rate of jujubes after the MCS and uses the pickup rate as a criterion for judging the pickup effect, the damage to jujubes in the device simulation test is ignored [31]. The parameters for the relevant materials in the designed simulation test are shown below (Table 1).

**Table 1.** Simulation contact parameter table.

| Parameters | Value |
|---|---|
| Poisson's ratio of sandy soil | 0.3 [19,32,33] |
| Sandy soil shear modulus/Pa | $1.15 \times 10^7$ [19,32,33] |
| Density of sandy soil/kg·m$^{-3}$ | 1379.07 [19,32,33] |
| Poisson's ratio of steel | 0.3 [19,33] |
| Steel shear modulus/Pa | $7.0 \times 10^7$ [19,33] |
| Steel density/kg·mm$^{-3}$ | 7850 [19,33] |
| Jujube Poisson's ratio | 0.248 [19,33] |

**Table 1.** *Cont.*

| Parameters | Value |
|---|---|
| Jujube Density/kg·m$^{-3}$ | 807.87 [19,33] |
| Shear modulus of jujubes/Pa | $6 \times 10^4$ [19,33] |
| Sandy soil-sandy soil Recovery Factor | 0.229 * |
| Sandy soil-sandy soil static friction coefficient | 0.609 * |
| Sandy soil-sandy soil rolling friction coefficient | 0.217 * |
| Sandy soil-steel recovery factor | 0.398 * |
| Sandy soil-steel static friction coefficient | 0.321 * |
| Sandy soil-steel rolling friction coefficient | 0.126 * |
| Jun jujube recovery factor | 0.35 [19] |
| Jun jujube-steel static friction coefficient | 0.309 [19] |
| Jun jujube-steel rolling friction coefficient | 0.035 [19] |
| Jun jujube-Jun jujube recovery factor | 0.25 [19] |
| Jun jujube-Jun jujube static friction coefficient | 0.48 [19] |
| Jun jujube-Jun jujube rolling friction coefficient | 0.04 [19] |
| Sandy soil-Jun jujube recovery factor | 0.33 [33] |
| Sandy soil-Jun jujube static friction coefficient | 0.79 [33] |
| Sandy soil-Jun jujube rolling friction coefficient | 0.16 [33] |

Explanation. "*" represents the data measured by the research group.

## 3. Results and Discussion

### 3.1. Analysis of Pickup Device Operation Parameters Based on DEM

#### 3.1.1. Simulation Model Establishment and Parameter Determination

According to the working principle of the pickup device designed and the above theoretical analysis presented in this paper, the key factors affecting the pickup rate are the depth of the comb teeth (CT) entering the soil (DCS), the CT working speed (CWS), the angle of the CT entering the soil (ACT), and the rotating speed of the strip brush roller shaft through the predict test (pretest). Among them, the speed of the brush roller shaft is affected by the CWS in the actual operation, which needs to reach a certain picking speed ratio, so the pickup effect is mainly affected by the CWS. At the same time, from the theoretical analysis of the previous section, the ACT has an impact on the pickup operation. However, through the pretest, it was found that the ACT can change freely within a certain range and has little impact on the picking operation, provided that the jujubes can be scooped up and the CTs have a certain ability to break the soil. Therefore, after determining the range of freely changeable ACT, it is only necessary to adjust the two parameters of the CWS and the DCS to meet the operation requirements.

Combining with the theoretical analysis and pretests, the DCS and the CWS were considered the main test factors affecting the jujube pickup rate. The research group found that a certain level of soil hilling quantity can help to improve the pickup rate of jujube in the field verification tests, so the simulation test uses the pickup rate of jujube and the soil hilling quantity on the surface of the CT as the test indicators. The parameter range of the DCS is 0–60 mm, and the DCS is nonzero in actual operation. Therefore, we set the DCS range to 5–65 mm, and the CWS was taken as 0.1–0.6 m/s in the simulation test [17,33]. According to the design calculation of the picking speed ratio of the pickup device, the picking requirement can be satisfied when the speed of the strip brush roller shaft is 65r·min-1. The simulation test was repeated three times, and the arithmetic average values were taken as the results.

#### 3.1.2. Simulation Test of ACT

In order to confirm the freely changeable range of the ACT, we analyzed the pickup rate of jujube and the influence rule of the soil hilling quantity under different ACT. In the simulation test, we set the ACT from 22° to 40°, and the test results are shown in Figure 9 [17].

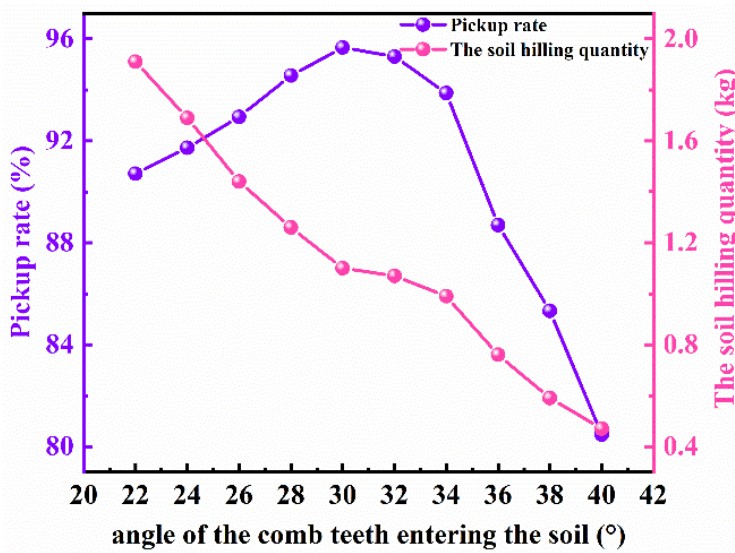

**Figure 9.** Variation curve of test index with the ACT.

According to references [2,5,6,17], the pickup rate of the existing pickup device is generally not less than 90%. Combined with the actual operational requirements, this paper uses whether the pickup rate reaches 92% as a reference standard for pickup effectiveness. According to Figure 9, when the ACT is less than 26°, the pickup effect of jujube is moderate, and it has some soil hilling quantity. The reason may be that, after the operation of the pickup device in the simulation test, the jujube is buried in the soil in the row spacing. In actual operation, a greater soil hilling quantity will bury the jujube and lead to the weakening of the pickup effect. When the ACT is greater than 35°, the pickup effect of jujube declines, and the amount of dammed soil on the comb surface reduces significantly. The reason may be that the increase in the angle weakens the ability of the CT to scoop up the jujube, and reducing the soil hilling quantity also makes jujube more difficult to accumulate, leading to a low pickup rate. When the ACT is 26–35°, it has a favorable influence on the pickup effect. The reason may be that when the angle of the CT is increased, jujube can still pile up along the CT and climb smoothly. With the decrease in sandy soil accumulation, the jujube will not fall back due to their excessively high accumulation on the CT surface. At the same time, the phenomenon of the sandy soil burying the jujube is reduced, and the strip brush can pick up more jujube, leading to an increase in the pickup rate. At the same time, combined with the simulation results, it is discovered that when the ACT is 28–33°, the change in angle has a comparatively small impact on the pickup effect of jujube, and the level of soil hilling quantity is not significantly different.

In actual production, the pickup device should not only assure the pickup effect, but also ensure that the CT can break the massive volume of soil. Nonetheless, with the increase in the angle of soil entry, the sandy soil disturbance of the CT to the sandy soil particles becomes stronger, which signifies that the ability of the CT to break the soil is strengthened. Next, we analyzed the accumulation effect and sandy soil disturbance of jujubes under different ACTs. When the ACT is 30–33°, the change in the jujube pickup effect and the soil hilling quantity is not obvious. However, under this condition, the disturbance of the CT to the sandy soil particles is evident, so the CTs have a strong ability to break the soil. Combined with the simulation test results shown in Figure 10a,b, we discovered that the optimal value of the change range of the ACT was 30–33°. This parameter range cannot only guarantee the pickup effect and the soil hilling quantity, but also provides considerable ability to break the massive volume of soil.

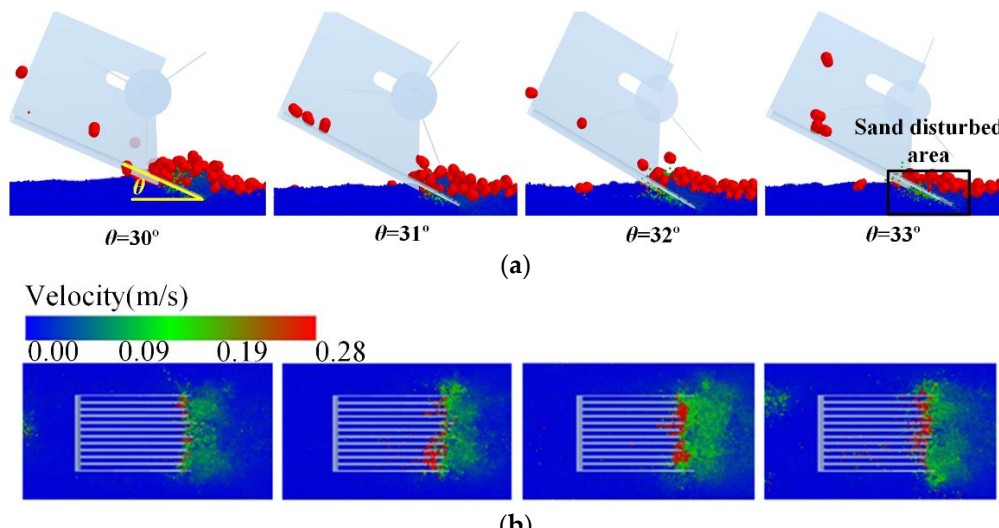

**Figure 10.** Schematic diagram of the basic conditions of jujubes and sandy soil under different ACTs. (**a**) Schematic diagram of the accumulation effect of jujubes and sandy soil under different ACTs. (**b**) Schematic diagram of the disturbance of sandy soil under different ACTs.

### 3.1.3. Analysis of a Single-Factor Simulation Test of CWS and DCS

In order to investigate the influence of the CWS and the DCS on the test index and determine the optimal value range of the working parameters, we conducted a single-factor simulation test of the pickup rate and the soil hilling quantity; the test results are shown in Figure 11.

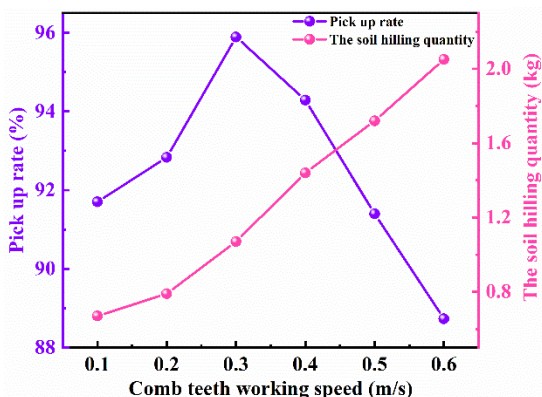

**Figure 11.** Schematic diagram of the effect of CWS on the test index.

Figure 11 shows the influence of CWS on the test index. When the CWS is less than 0.3 m/s, the pickup rate of jujube increases almost linearly, and the soil hilling quantity also increases with the increase in the speed. The reason may be that the increase in speed accelerates the accumulation of sandy soil and jujube. The structure formed by accumulation is relatively stable and moderately high. The jujube in this state is easily picked up by the strip brush. As the speed of sandy soil accumulation accelerates, the soil hilling quantity increases. When the CWS is 0.3–0.6 m/s, the pickup rate decreases with the increase in speed, but the soil hilling quantity continues to increase. The reason is that the jujube is raised too high during this process; although the strip brush easily picks it up, the accumulation structure of the jujube becomes unstable and easily dispersed. At the same time, the jujube is buried by the sandy soil, leading to a decline in the pickup rate. Another possible reason for the decrease in pickup rate may be related to the width of the MCS. In actual operation, the width of the MCS is less than 1000 mm, mainly between 600 and 1000 mm. The simulation model of the pickup device in this study has a working width of

340 mm and an MCS width of 220 mm. At a faster working speed, piled-up jujubes quickly fall back, and jujubes at the edge of the row spacing will be dispersed directly outside the pickup device, resulting in a decrease in pickup rate. The CT working speed range of combs of 0.2~0.4 m/s was selected through the above analysis combined with actual operation requirements and the standard pickup effect in this paper.

To observe the scattering rule of jujube under different CT speeds more precisely, and to provide a basis for future machine design, in this paper, under the conditions of different CWS, the vector map of jujubes is drawn under the simulation of the obvious scattering phenomenon of jujubes (as shown in Figure 12; CWS are 0.1 m/s, 0.5 m/s, and 0.6 m/s, respectively). It can be seen from the scattering law of jujube in Figure 12 that, when the CWS is 0.1 m/s, most jujube scatter in comb area II, a few jujube scatter from both sides, and the sandy soil particles' disturbance is not obvious. The reason may be that the CWS is slow. Sandy soil does not easily accumulate and can mostly be filtered through the gap between the CTs. The accumulation speed of jujube is also slow. The scattering of jujube becomes stronger without the support of the sandy soil. Therefore, most of the jujubes fall back in the middle, and the jujubes will not be scattered from both sides of the CT with the sandy soil. At a speed of 0.5–0.6 m/s, the accumulation speed of sandy soil and jujube is fast, and the accumulation height of sandy soil and jujube is increased more quickly. The strip brush rapidly picks up the materials at the top of the accumulation. However, due to the fast CWS and the rapid increase in sand accumulation, the phenomenon of jujubes scattering from both sides of the CT is more significant, which means that some jujubes are scattered from both sides of the CT along with the sandy soil, thus, being buried by the sandy soil. The speed range of CT was determined as 0.2~0.4 m/s, according to the vector illustration of jujube.

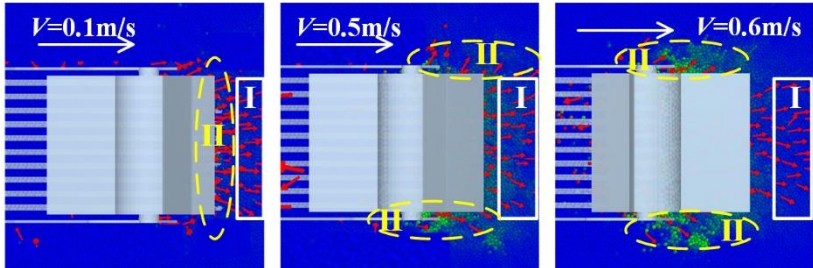

I is jujubes that have not been picked up; II is the scattered jujubes

**Figure 12.** Vector image of red dates under different CWS.

Figure 13 presents the curve of the test index changing with the DCS. When the DCS is less than 25 mm, the pickup rate of jujube increases with the DCS, and the soil hilling quantity increases accordingly. When the DCS is more than 25 mm, the pickup rate decreases with the increase in the DCS, but the soil hilling quantity increases continuously. From the judgment criterion of the pickup effect, the pickup effect is better at a DCS of 15–45 mm, and the soil hilling quantity increases continuously with the increase in the DCS. In this state, the increase in the soil hilling quantity will play a positive role in the pickup effect, which also confirms the results of the pretest of the subject team. When the DCS is 25~45 mm, the pickup effect is better, but the pickup rate decreases. When the penetration depth of the comb is greater than 45 mm, the pickup effect is poor, and the backup volume increases continuously with the decrease in pickup rate. Moreover, due to the continuous increase in the soil hilling quantity, a large amount of sandy soil is scattered from both sides of the CT, resulting in a more significant phenomenon of jujubes being scattered from both sides along with the sandy soil. In practice, when the soil hilling quantity is large, the strip brush will be blocked by the soil hilling and bent, which results in a poor pickup effect. At the same time, the accumulation of jujube branches, leaves, and jujubes after MCS will also lead to a greater height of the accumulation body in the field than in the simulation, making

the picking operation more difficult. Therefore, excessive soil hilling quantity during the operation will have a strongly negative impact on the pickup operation.

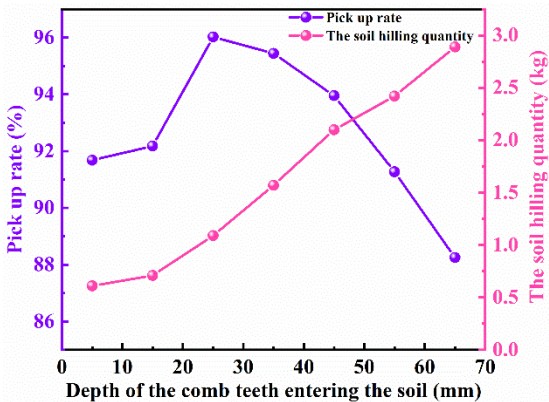

**Figure 13.** Influence of different DCS on test index.

We sought to better analyze the pickup effect under different DCS and determine a reasonable range of parameter values. Figure 14 shows the simulation test results of the DCS at 5 mm, 15 mm, 25 mm, 35 mm, 45 mm, 55 mm, and 65 mm, respectively. It can be seen from Figure 14 that when the DCS is 15–35 mm, the stacking state of jujubes is good. Among them, when the DCS is 15 mm, the sandy soil accumulation is relatively low and the jujube accumulation is relatively high. However, due to the small amount of sandy soil supporting the jujube during the operation, the stability of the structure of the accumulation body is poor, leading to the phenomenon wherein the jujube is scattered back to the row spacing. When the DCS is 45~65 mm, the phenomenon of sand being swept up by the strip brush is significant, which also indicates that the soil hilling quantity increases obviously. An excessive soil hilling quantity will hinder the normal operation of CT and cause problems, such as burying jujubes. Therefore, according to the actual topographic conditions of the jujube orchard [17], DCS is determined to be 5–45 mm when combining the analysis of the pickup effect.

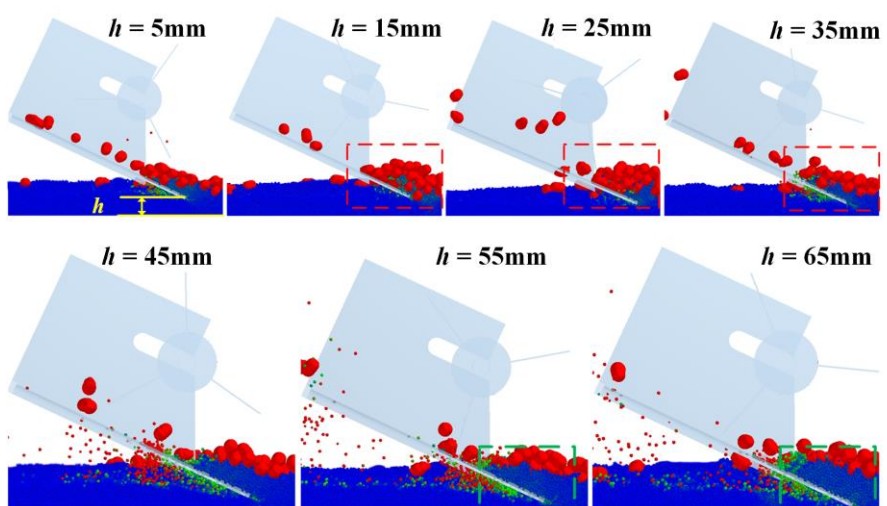

**Figure 14.** Simulation effect under different DCS.

### 3.2. Field Test Materials

In order to verify the reliability of the simulation test and the feasibility of the selected parameter range, in the simulation test and field verification test, the test factors are set as the CWS and the DCS, and the test index is set as the pickup rate of jujubes. According to a study of the ACT, the ACT is not used as the test factor. The field verification tests of each

group were repeated three times, and the results were taken as the average values. The field test was conducted in the trunk jujube demonstration base of Kunlun Mountain Jujube Industry Co., Ltd., of the 224th regiment of the 14th division of the Xinjiang Production and Construction Corps. The soil type was sandy soil, and the soil moisture content was 10.12–15.73%. The fruit was a 4-year-old Jun jujube. The jujube garden was 4 × 1.5 m (row spacing × plant spacing). The test equipment included a pickup device, a tape measure, an ACS-30D electronic scale, an inverter, a fan, etc.

### 3.3. Field Test Methods

The test referred to the standard of the DG-T 188-2019 fruit picker [34], and several test areas with a length of 20 m were distinguished along the rows of jujube trees, while the fan was used to manually gather the ground jujubes. During the test, the pickup device was built on the floor-standing jujube pickup machine developed by the research group. We adjusted the operating parameters of the ground jujube pickup device to meet the test requirements and drove into the test area for tests (Figure 15). The test index picking rate refers to the ratio of the picked jujube to the total mass of the jujube after MCS in the test area. The calculation formula is as follows:

$$Y_{1\text{-}1} = \frac{M_{1\text{-}1}}{M_{2\text{-}1}} \times 100\% \tag{20}$$

where $Y_{1\text{-}1}$ is the pickup rate of jujubes, %; $M_{1\text{-}1}$ is the mass of picked jujubes, kg; $M_{2\text{-}1}$ is the total mass of jujube after MCS in the test area, kg.

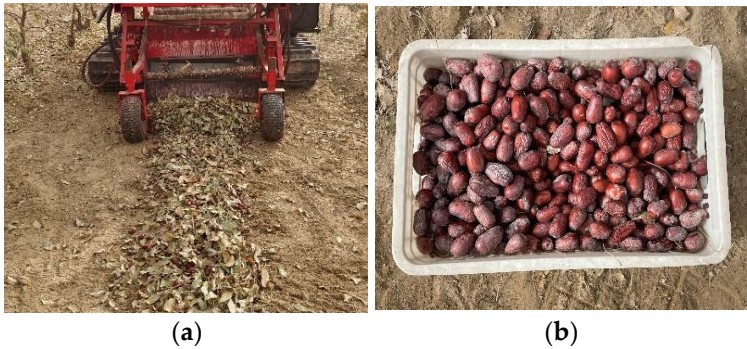

|         (a)         |         (b)         |

**Figure 15.** (**a**) Field test of a floor-standing jujube pickup device. (**b**) Jujubes picked up by devices.

### 3.4. Analysis of Field Test Results

According to the test methods and standards formulated above, the relevant test results are shown in Table 2.

**Table 2.** Test results.

| Test Sequence | CWS/ (m/s) | DCS/mm | Simulation Test Pickup Rate/% | Filed Test-Pickup Rate/% | Relative Error/% |
|:---:|:---:|:---:|:---:|:---:|:---:|
| 1 | 0.2 | 5 | 84.17 | 91.23 | 7.73 |
| 2 | 0.2 | 25 | 94.31 | 93.49 | 0.08 |
| 3 | 0.2 | 45 | 94.97 | 94.52 | 0.47 |
| 4 | 0.3 | 5 | 87.53 | 92.66 | 5.53 |
| 5 | 0.3 | 25 | 96.61 | 95.03 | 1.6 |
| 6 | 0.3 | 45 | 94.34 | 93.69 | 0.69 |
| 7 | 0.4 | 5 | 85.77 | 90.27 | 4.98 |
| 8 | 0.4 | 25 | 94.29 | 94.72 | 0.45 |
| 9 | 0.4 | 45 | 92.82 | 91.78 | 1.13 |

From the field test results, it can be seen that the test error is the largest when the CWS is 0.2 m/s, 0.3 m/s, or 0.4 m/s, and the DCS is 5 mm. Since the jujube after MCS in

the simulation test does not contain jujube branches, jujube leaves, or other materials, the fluidity of the sandy soil particles is good. However, in the actual operation, the jujube belt after MCS contains jujube branches, jujube leaves, and other materials, resulting in the structure formed by sandy soil accumulation being more stable than that in the simulation test, and the fluidity of the sandy soil particles becoming poorer. Moreover, the actual stacking height ratio will also be higher than that of the simulation test, so the strip brush can more easily pick up fruit. Therefore, under these conditions, the pickup rate in the field test is significantly higher than that in the simulation test. When the CWS is 0.2 m/s, 0.3 m/s, or 0.4 m/s, and the DCS is 45 mm, the pickup rate in the field test is lower than that in the simulation test. The reason may be that the field test environment is complex. The deeper the soil layer, the more impurities are contained in the actual operation, which will reduce the fluidity of the sandy soil, which will in turn make it difficult for the sandy soil to fall from the CT clearance, and most of it will fall from the top or both sides of the accumulation, resulting in a more serious phenomenon of jujube burial. Therefore, the pickup rate is low compared with the simulation test. However, the overall error of the field verification experiment and simulation experiment is less than 8%, which proves that the simulation experiment scheme designed in this paper is relatively reliable and that the range of test parameters determined is feasible.

*3.5. Discussion*

The results show that when the CWS is constant, the acceleration force $F_b$ of the soil is also constant. The gravity of soil block $G$ is affected by the depth of the CT. The greater the depth at which the CT enters the soil, the greater the gravity of the soil, the more soil will accumulate along the CT, the more likely the dates will be buried by the soil, and a certain amount of congestion will occur. Therefore, the main factors affecting the working state are the CWS, the DCS, and the ACT. This is consistent with the research results in the literature [12,14,17] et al. Therefore, the EDEM simulation model established in this paper can be used to predict the operation performance of the floor-standing jujube pickup device in a working process. The research can provide important theoretical and methodological support for designing the operating parameters of agricultural machinery equipment.

In addition, the pickup rate of the floor-standing jujube picker developed by the literature [33] et al. is 90%, and the pickup rate in this study reached 92%. The pneumatic jujube harvester described in the literature [34] has a pickup rate of 98%, which is one of the highest pickup rates for machines. However, the lower working efficiency cannot satisfactorily fulfill the production demand. According to the comprehensive analysis, the reason is that the technology of jujube pickers is not mature at present, and the mechanical jujube picker is limited by the working principle and working environment; thus, the pickup rate is low, but the working efficiency is high. The pneumatic date harvester has a high pickup rate, but there are high requirements in terms of the jujube orchard soil type and land leveling. In addition, according to the simulation test and field test results, the research group found that the pickup rate was also greatly affected by regional factors, such as the size of jujube branches and leaves, soil type, and soil moisture content.

## 4. Conclusions

In this paper, according to the operating characteristics of the floor-standing jujube pickup device, the structural parameters of the comb teeth (CT) and the strip brush in the pickup device are determined through theoretical analysis. The model of the virtual simulation test was established using EDEM software, and the optimal free variation range of the angle of the CT entering the soil (ACT) was determined to be 30–33° through a pre-test and simulation analysis. Taking the CT working speed (CWS) and the ACT as the test factors, single-factor tests were carried out on the pickup rate and the soil hilling quantity. Through the tests, it was determined that the optimal parameter range of the CWS was 0.2~0.4 m/s, and the value range of the depth of the CT entering the soil (DCS) was 5~45 mm. The field validation test was carried out through the determined parameter range,

and the maximum error of the test was not more than 8%, which proved that the designed simulation test was more reliable, and that the determined operating parameter range was more feasible. This work can provide a methodological reference and a parametric basis for analyzing the soil disturbance patterns of jujube pickup devices.

In addition, in the future, attention should be paid to research on the breakage rate of jujubes caused by pickup devices.

**Author Contributions:** Conceptualization, methodology, data curation, formal analysis, writing—review and editing, validation, writing—original draft preparation, L.Z. and J.L. (Jingbin Li); investigation, J.L. (Junpeng Liang); data curation, L.D.; funding acquisition, J.L. (Jingbin Li) and L.D.; validation, H.D.; supervision, J.L. (Junpeng Liang). All authors have read and agreed to the published version of the manuscript.

**Funding:** This research was funded by the Regional Innovation Guidance Plan of the Xinjiang Production and Construction Corps (2021BB003), the National Nature Science Foundation of China (51865050), and the project to improve the level of agricultural science and technology of the Xinjiang Production and Construction Corps (2130106-13).

**Institutional Review Board Statement:** Not applicable.

**Data Availability Statement:** All data are presented in this article in the form of figures and tables.

**Conflicts of Interest:** The authors declare no conflict of interest.

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
