# Peer review of "Analysis and Design of Operating Parameters of Floor-Standing Jujube Pickup Device Based on Discrete Element Method"

_agriculture, doi:10.3390/agriculture12111904_

Round 1

Reviewer 1 Report

1. Whether there is any error in the relation in Equation (15), combined with Figure 6a and Equation (16), it should be ..

2. The caption in Table 1 is wrong. It should be Table 1.

3. The CWS in Table 2 are 0.2m/s, 0.3m/s and 0.4m/s, respectively, but the following analyses are 0.2m/s, 0.4m/s and 0.6m/s.

4. It can be seen from Figure 8 that 10 comb teeth are used in the simulation experiment. If 37 comb teeth are needed for the working width of 1100mm as mentioned in Section 2.3.4, it seems that the 10 comb teeth are only enough for the working width of 300mm. Is there any wrong marks in the figure?

5. References in Section 3.1 are not annotated completely.

Reviewer 2 Report

The following comments were obtained after peer-reviewing the article:

Analysis and design of operating parameters of floor-standing jujube pickup device based on discrete element method.

I found this work excellent.

Main comments

In the introduction of the paper a paragraph should be included to highlight the importance of the crop, its main use and harvesting season.

In the material and method section it should bring a clear paragraph of what the authors are going to do. It is clear how the mathematical analysis is done. However, the simulation and comparison against an operating machine are the results of the work. This work is excellent but I would change the simulation to the result section starting at section 2.5.

In figure 2a and figure 2 b headings I recommend to substitute each number 3. Or 4. By (3) and (4) as it it easier to read.

The angle of formulas 2 has a different notation in Figure 4 so it has to be fixed.

In Figure 5, the letters within the drawing cannot be seen clearly. I suggest to change the color blue by a brown clear color proper of sand. It will let a clearer view of the letters and contrast with the jujube.

In line 226 the unit within () can be eliminated as there is no other angle unit.

The two equations number 11 (line 232) calculate Wg and Wf but they their meaning is not mentioned in the text.

Please move part of the paragraph starting at line 307 to the discussion section.

In line 364, once the simulation has began it will be important to rename in the text what is DCS, CWS and ACT.

I am confused by the term pick up rate, as it will have a relationship with the speed of picking. As figures 9 and 11 vary in %, I will suggest to use in text and graphs pick up efficiency.

The paragraph starting in line 398 discusses the reason why variations are found and it should be in your discussion section.

In Figure 10 b the value of each speed should be over each image unless the color represents the speed.

The sentence between lines 441 and 444 is not clear having too many sand terms. Lines 438-440 is exactly the same as line 446-447.

In Fig 15, the heading has to be fixed having (a) and (b) continuously.

All the authors mentioned in the discussion section from lines 575 to 579 have to be removed letting only the reference number.

References:

The first two references in lines 620-623 do not present year and page numbers.

Reference 12 in lines 641 and 642 lack volume and page numbers.

References 8, 9 and 21 miss commas after each author.

Reference 23 is not well described as it has a + in the pages’ number.

Reference 21 is a USDA book referred as Agriculture Handbook 316 but it is not included in the text.
